# A comparison of educational events for physicians and nurses in Australia sponsored by opioid manufacturers

Quinn Grundy[1,2]*, Sasha Mazzarello[1], Sarah Brennenstuhl[1], Emily A. Karanges[2,3,4]

**1** Lawrence S. Bloomberg Faculty of Nursing, University of Toronto, Toronto, Canada, **2** Faculty of Medicine and Health, Sydney Pharmacy School, The University of Sydney, Sydney, Australia, **3** Faculty of Medicine, Dentistry and Health, Centre for Youth Mental Health, The University of Melbourne, Melbourne, Australia, **4** Orygen, Melbourne, Australia

* quinn.grundy@utoronto.ca

**Data Availability Statement:** The full dataset is publicly available at https://researchdata.ands.org.au/pharmaceutical-industry-funded-sept-2015/941218.

## Abstract

### Background

Educational activities for physicians sponsored by opioid manufacturers are implicated in the over- and mis-prescribing of opioids. However, the implications of promotion to nurses are poorly understood. Nurses play a key role in assessing pain, addressing the determinants of pain, and administering opioid medications. We sought to understand the nature and content of pain-related educational events sponsored by opioid manufacturers and to compare events targeting physicians and nurses.

### Methods

We conducted a cross sectional, descriptive analysis of pharmaceutical company reports detailing 116,845 sponsored educational events attended by health professionals from 2011 to 2015 in Australia. We included events that were sponsored by manufacturers of prescription opioid analgesics and were pain related. We compared event characteristics across three attendee groups: (a) physicians only; (b) at least one nurse in attendance; and (c) nurses only. We coded the unstructured data using iteratively generated keywords for variables related to location, format, and content focus.

### Results

We identified 3,411 pain-related events sponsored by 3 companies: bioCSL/CSL (n = 15), Janssen (n = 134); and Mundipharma (n = 3,262). Pain-related events were most often multidisciplinary, including at least one nurse (1,964/3,411; 58%); 38% (1,281/3,411) included physicians only, and 5% (166/3,411) nurses only. The majority of events were held in clinical settings (61%) and 43% took the form of a journal club. Chronic pain was the most common event topic (26%) followed by cancer pain and palliative care (18%), and then generic or unspecified references to pain (15%); nearly a third (32%) of event descriptions contained insufficient information to determine the content focus. Nurse-only events were less frequently held in clinical settings (32%; p < .001) and more frequently were product launches

**Funding:** This study was funded by a Connaught New Researcher Award from the University of Toronto to QG. The funders had no role in study design, data collection and analysis, decision to publish, or preparation of the manuscript.

**Competing interests:** The authors have no conflicts of interest. Quinn Grundy currently serves as Academic Editor for PLOS ONE. This does not alter our adherence to PLOS ONE policies on sharing data and materials.

(17%; p < .001) and a significantly larger proportion focused on cancer or palliative care (33%; p < .001), generic pain topics (27%; p < .001), and geriatrics (25%; p < .001) than physician-only or multidisciplinary events.

## Discussion

Opioid promotion via sponsored educational events extends beyond physicians to multidisciplinary teams and specifically, nurses. Despite lack of evidence that opioids improve outcomes for long-term chronic non-cancer pain, hundreds of sponsored educational events focused on chronic pain. Regulators should consider the validity of distinguishing between pharmaceutical companies' "promotional" and "non-promotional" activities.

## Introduction

Opioid-related overdoses and deaths constitute a contemporary public health crisis in several countries including the United States, Canada and Australia. While the United States and Canada continue to lead in terms of prescribing and utilisation of opioid analgesics [1], and the burden of opioid-related mortality [2], Australia appears to be following a similar pattern [3]. Between 1992 and 2012, the dispensing of opioid medications in Australia increased 15-fold, marked by an increase in the use of strong, long-acting and transdermal formulations [4]; specifically, there was a major shift from the use of dextropropoxyphene and morphine in 1990, to oxycodone, tramadol and fentanyl in 2014 [5]. During the same time period, opioid-related harms also increased including opioid-related hospitalizations and deaths due to accidental poisoning [4, 6], which were attributed more often to prescription opioids than to heroin [6].

Through the 1990s and 2000s, opioid manufacturers aggressively promoted and heavily marketed opioid medications, particularly for chronic, non-cancer pain, despite a paucity of evidence on their safety and efficacy for long-term use [7, 8]. For example, Purdue Pharma introduced a sustained-release oxycodone preparation (OxyContin) in the United States in 1996, which quickly reached blockbuster status with combined sales of nearly $3 billion (over 14 million prescriptions) by 2002 [7]. Compared with other available opioid preparations, Purdue Pharma's formulation offered no clinical benefits; thus, the explosion in sales of oxycodone was attributed to the company's aggressive promotional and marketing campaign [7]. Since 2010, prescriptions for OxyContin began to fall in the United States; however, the owners of Purdue Pharma, the Sackler family, pursued similar marketing strategies internationally through a global network of companies, Mundipharma [9]. In Australia, in 2016–2017, oxycodone (marketed by Mundipharma Australia) accounted for 37% of the 15.4 million opioid prescriptions dispensed [6].

However, there is relatively little research on the nature of opioid promotion outside of the United States. Journalists and researchers have primarily relied on industry documents made public through litigation (primarily in the United States) or databases of pharmaceutical industry payments to health professionals (such as the United States Open Payments database) to study the association between industry payments and opioid-related health outcomes [10–12].

In 2007, Australia was one of the first countries to move toward greater transparency of the relationships between health professionals and the pharmaceutical industry [13]. From October 2011 to September 2015, Medicines Australia, the trade association for the prescription medicines industry in Australia, required member companies to report the details of

sponsored events for all registered health professionals [14]. The dataset, which is publicly available (http://dx.doi.org/10.4227/11/592631edbd9d5), is unique globally in that it details sponsored educational events including all registered health professionals, whether or not they have prescribing authority, and, in addition to the dollar value of the sponsorship, contains a description of the event's content, the hospitality provided, the venue, and the number and professional status of attendees. Thus, we aimed to analyse the nature of pain-related educational events sponsored by opioid manufacturers to understand patterns in relation to the content and setting, and to compare events targeting physicians and nurses. While this analysis can provide historical insight into patterns of opioid prescribing and use in Australia, it also provides a snapshot of the ways that health professional education may serve as a facet of drug promotion more broadly.

## Education as promotion

The resulting legal evidence stemming from United States litigation of brand name opioid manufacturers suggests that pharmaceutical industry promotion to health professionals was a key factor in seeding and exacerbating the epidemic of opioid-related overdose and mortality. Primarily, promotional and educational campaigns misrepresented and greatly downplayed the risk of addiction [7, 15]. Regulators have taken note: in Canada in June 2018, Health Canada called on opioid manufacturers and distributors to voluntarily stop all promotional activities and advertising to health professionals [16].

While there has been scrutiny of pharmaceutical sales practices, including large bonuses for representatives and coupons for free samples, opioid manufacturers' involvement in medical and continuing education has also been flagged as another means by which companies promoted the over-prescribing of opioids [7, 8, 17]. Analysis of internal industry documents suggests that a key pharmaceutical industry marketing strategy is coordinated "education" campaigns comprised of a number of activities not typically recognized as 'promotional' including use of key opinions leaders, peer selling, and industry sponsorship of continuing education [18, 19]. Purdue Pharma's promotion of a sustained-release oxycodone formulation included more than 40 national conferences attended by over 5000 health professionals, and sponsorship of more than 20,000 pain-related educational programs [7]. A case study of medical education at a research-intensive university found that an annual interprofessional pain curriculum, sponsored by pharmaceutical companies including opioid manufacturers, included pharmacotherapy lectures and free textbooks that emphasized the effectiveness of opioids for chronic non-cancer pain and minimized the risks of harm and addiction [15]. These lectures were delivered by physicians who received payments from opioid manufacturers, which was not always disclosed to students [15].

Despite being characterized as "educational," industry-sponsored events are almost always associated with other payments and/or gifts: food and drink were provided at over 90% of pharmaceutical-industry sponsored educational events for health professionals in Australia between 2011 and 2015 [14]. Between 2013–2015, 1 in 12 United States physicians (and 1 in 5 primary care physicians) received a payment from an opioid manufacturer, which was most commonly in the form of food or beverages or speakers' fees, suggesting that attendance at industry-sponsored drug dinners or other educational events is widespread [10].

Analyses of United States data suggest that this kind of sponsored education and related payments such as speakers' fees, conference travel sponsorship, consultancies, and meals are associated with the increased and harmful prescribing of opioids. Researchers found that 7% of physicians who prescribed opioids under a national drug insurance program (Medicare Part D) received payments from opioid manufacturers; these physicians submitted 9.3% more

opioid claims the following year compared with physicians who did not receive payments [11]. Analysing the impact of these payments at the county level per 1000 county population, researchers found that increases in the number of payments to physicians, the number of physicians receiving marketing, and the total monetary value of payments to physicians were associated with increased morality from opioid overdoses [12].

### Opioid promotion to nurses

The focus of public, regulatory, legal, and academic scrutiny of opioid-related promotion has largely remained on promotion to physicians [10–12], and to some extent nurse prescribers [20]. In the United States, a nurse prescriber plead guilty to accepting $83,000 in kickbacks in exchange for prescribing an opioid medication, which she accepted in the form of speaking engagements at educational events that were often attended only by the company's sales representatives or friends without prescribing authority [20]. However, the implications of opioid promotion to registered nurses without prescribing authority are poorly understood.

Nurses, which comprise the largest proportion of health professionals across health systems, frequently interact with industry representatives and also report receiving payments and gifts [21]. In Australia, 40% of pharmaceutical-industry sponsored educational events included nurses; in contrast, 21% of sponsored events included primary care physicians [22]. Through promotional campaigns, pharmaceutical companies seek not only to increase their market share, but also to expand their market through promoting awareness, screening, assessment of conditions for which the drug is indicated and expanding the indications for the drug, particularly to new settings and populations [23] Nurses play a key role in assessing pain, and administering opioid medications and particularly, medications prescribed 'as needed,' across multiple healthcare settings including acute care, primary care, and long term care. Nursing practice also attends to the multiple psychosocial factors that serve as determinants of pain and they play a large role in coordinating care, referrals, and the administration of alternative modalities for pain treatment [24, 25]. Thus, it is important to understand patterns in promotion targeted at nurses as well as physicians, particularly as nurses' relationships with industry have received less policy attention than physicians'. In Australia, nurses' relationships with industry are primarily governed by the industry Code of Conduct [26], and there is little guidance specifically for nurses from professional associations, the health system or the government. To date, there has been no investigation into the promotion of opioids specifically to nurses, thus we aimed to compare the nature of pain-related educational events sponsored by opioid manufacturers that were targeted at physicians and nurses.

## Methods

### Design

We conducted a cross-sectional content analysis of a public database of pharmaceutical industry-sponsored educational events for health professionals [14]. This study was exempt from ethical review according to the guidelines of the University of Toronto.

### Setting

From October 2011 to September 2015, Medicines Australia required member companies to report the details of all industry-sponsored events for health professionals, defined as any registered health professional, and including registered nurses; companies posted PDF reports every 6 months on the Medicines Australia website. The database was created by Fabbri et al [14] who downloaded and compiled 301 separate PDFs from 42 member companies,

converted these to Excel format, and cleaned the data for errors introduced in file conversion and to standardize the reporting. This dataset provides a unique case study to examine the nature of pharmaceutical industry-sponsorship of opioid- and pain-related events and particularly, those that include nurses.

In 2015, 12 opioid analgesics were available in Australia; 9 of these were also approved for subsidy by the Australian Government under the Pharmaceutical Benefits Scheme, including buprenorphine, codeine, fentanyl, hydromorphone, methadone, morphine, oxycodone, tapentadol and tramadol [27].

## Sampling

Industry-sponsored educational events were included if they were sponsored by manufacturers of brand name prescription opioid analgesics approved by the Therapeutic Goods Administration (TGA) from 1996 to present, who are also member companies of Medicines Australia (Table 1). This time frame ensured that the product was likely on patent during the study time frame (Oct 2011 to Sept 2015). Thus, the inclusion criteria for events were sponsorship by a company:

- With Medicines Australia membership during the study period (Oct 2011 to Sept 2015)

- That manufactured a prescription opioid approved by the TGA from 1996 to present and whose primary indication was for analgesia

   Exclusion criteria included:

- Sponsor not a member of Medicines Australia during the study period (e.g. generics manufacturers, opted out of membership)

- Product approved by the TGA prior to 1996 (e.g. codeine, codeine+paracetamol, pethidine, methadone, dextropropoxyphene)

- Product not on the market by September 2018

- Manufacturers of opioids whose primary indication is anaesthesia, substance use disorder treatment, or breakthrough cancer pain as these products have very low usage in comparison with other opioids [5]; in all cases, companies that manufactured opioids for these indications marketed only one product and sponsored few to no pain-related events

- Manufacturers of medicines for hospital-use only

   Thus, we included all events sponsored by:

- Janssen-Cilag Pty Ltd: a subsidiary of Janssen Global, the pharmaceutical branch of Johnson & Johnson, manufactured pharmaceutical products in the areas of mental health, neurology, women health, haematology, gastroenterology, and pain management.

- Mundipharma Pty Ltd: founded in 1998, Mundipharma is a privately held company owned by the Sackler family (of Purdue Pharma) and part of the global network of Mundipharma companies, which marketed prescription drugs primarily in the area of pain management, as well as an asthma inhaler and oncology symptom management.

- bioCSL / Seqirus Australia Pty Ltd: a member of the CSL Group, a global specialty biotechnology company produced a range of vaccines, antivenoms, and pharmaceuticals related to pain management and allergies; the company is now known as Seqirus.

**Table 1. Prescription brand name opioid medicines approved by the TGA from 1996 to September 2018.**

| Generic name | Brand name | Formulation | Date approved (TGA) | Date listed (PBS)[a] | Indication[c] | Company | Include |
|---|---|---|---|---|---|---|---|
| Buprenorphine | Norspan | Transdermal patch | 2005/05 | 2005/12 | Chronic severe pain | Mundipharma | Y |
| Fentanyl | Durogesic | Transdermal patch | 1997/10 | 1999/08 | Chronic cancer pain Chronic severe pain | Janssen | Y |
| | | | | 2006/08 | | | |
| | Abstral | Lozenges | 2002/11 | 2008/04 | Palliative care (breakthrough cancer pain) | Menarini Australia | N |
| | PecFent | Nasal spray | 2012/08 | Application rejected | Breakthrough cancer pain | AstraZeneca | N |
| | Instanyl | Nasal spray | 2013/06 | Application rejected | Breakthrough cancer pain | Takeda | N |
| Hydromorphone | Dilaudid | Liquid | 1999/08 | 2000/08 | Severe disabling pain | Mundipharma | Y |
| | | Injection | | 2000/08 | Unrestricted benefit | | |
| | | SR tablets | | 2001/11 | Severe disabling pain | | |
| | Dilaudid-HP | Injection | | 2000/08 | Unrestricted benefit | | |
| | Jurnista | CR tablets | 2008/07 | 2009/05 | Chronic severe pain | Janssen | Y |
| Morphine | Kapanol | SR capsule | 1992 | 1994/12 | | GlaxoSmithKline | N |
| | MS Contin | CR tablets | 1991/08 | 1996/02 | Chronic severe pain (up to 120mg/tab) | Mundipharma | N |
| | | Suspension | | 1997/08 | Chronic severe pain due to cancer Chronic severe pain (palliative care) (200mg) | | |
| | Ordine | Oral solution | 1991/07 | 1994/06 | Chronic severe pain due to cancer | Mundipharma | N |
| | MS Mono | CR capsule | 2000/09 | 2001/02 | Chronic severe pain | Mundipharma | Y |
| | Sevredol | Tablet | 1994/04 | 2006/08 | Severe disabling pain due to cancer Severe disabling pain (palliative care) | Mundipharma | N |
| Oxycodone | OxyNorm | Capsules | | 2001/05 | Severe disabling pain | Mundipharma | Y |
| | | Liquid | | 2003/08 | | | |
| | OxyContin[b] | CR tablets | 1999 | 2000/05 | Chronic severe pain | Mundipharma | Y |
| Oxycodone +naloxone | Targin | CR tablets | 2010/05 | 2011/12 | Chronic severe pain | Mundipharma | Y |
| Tapentadol | Palexia | MR tablet | 2011/01 | 2014/06 | Chronic severe pain | bioCSL | Y |
| Tramadol | Tramal | Capsule | 1998 | 2000 | Moderate to severe pain | bioCSL | Y |
| | Tramal | Oral drops | | 2005/08 | Pain not responding to aspirin and/or paracetamol | | |
| | Tramal SR | SR tablet | | 2006/12 | Acute pain not responding to aspirin and/or paracetamol | | |
| | Durotram XR | SR tablet | 2008 | Delisted in 2013 | Moderate to severe pain | iNOVA | N |

CR = controlled release; MR = modified release; SR = standard release

[a]Taken from PBS Public Summary Document for each medicine

[b]Note: modified release tablets PBS-listed for chronic non-cancer pain (2000); replaced with tamper-resistant tablet 2014/04/01; non-tamper proof controlled-release tablet withdrawn; 2014/09/01 release of generic slow-release deterrent (not subsidised)

[c]Pharmaceutical Benefits Scheme (PBS) listings of opioid analgesics as of 1 July 2014: https://www.pbs.gov.au/industry/listing/participants/public-release-docs/opioids/opioids-dusc-prd-2014-10-final.pdf

## Coding

We coded all events using an iteratively generated keyword search strategy that expanded the coding strategy established by Fabbri et al [14]. Fabbri et al [14] generated keywords based on theoretical variables of interest as outlined in the literature on industry-health professional

interactions (e.g. clinical specialty, clinical setting, hospitality provided) and then iteratively expanded the set of keywords within each variable to account for the nature and range of events in the dataset (see S1 Table). We developed a coding scheme for pain-related content, which was first piloted by two independent coders on a random sample of pain-related events and refined among the team. Where information was missing in the "description of event" column, events were coded using keyword searches in the "venue" and "professional status" columns where content was unambiguous. We coded all Mundipharma events as "pain-related" unless the event explicitly stated otherwise, given the company's limited product profile and the dominance of opioid medications within this portfolio. We further assumed that events were "pain-related" in the absence of details about the event content (e.g. 1 hour Journal club) given that Mundipharma is a member of the global network of Mundipharma companies (which are 100% owned by the Sackler family, the owners of Purdue Pharma) and Mundipharma's aggressive promotion of opioid medications and particularly oxycodone internationally is well-documented [9]. We conducted additional keyword searches to exclude events as "not pain-related" based on Mundipharma's non-opioid product offerings, which at the time of the study, included respiratory products for asthma, diabetes products, and oncology products related to lymphoma and anti-nausea agents (see S1 Table).

We coded all events for professional status (i.e. physician, nurse, etc) as 'present' or 'absent' using a keyword search in the "professional status" column. We then coded events into five mutually exclusive categories (see S1 Table):

1. Physician-only

2. At least one nurse in attendance

3. Nurses only

4. Other health professionals only

5. Other health professionals and physicians only

We elected not to analyse events for nurses with and without prescriptive authority separately, inferring that non-prescribing nurses are routinely included in pharmaceutical industry-sponsored events in Australia. In Australia, Nurse Practitioners (NPs) are a sub-category of registered nurses (RNs) who have prescriptive authority. In a previous analysis of transparency reports detailing payments from Australian pharmaceutical companies to individual health professionals, we found that companies did not always distinguish between RNs and NPs, referring to individuals as "nurses" [28]. However, nurses with an NP license likely account for a minority of nurses in receiving industry payments or attending sponsored events. Though NPs in Australia have a similar scope of practice to NPs in the United States (US), NPs in Australia are much fewer in number (1,319 in Australia vs 136,060 in US) and represent a smaller proportion of all RNs (representing about 0.005% of all Australian RNs vs about 5% of US RNs) [29]. From September 1st 2010, nurse practitioners were authorised to prescribe the opioid medications under study, though their prescribing was limited by scope of practice, state and territory prescribing rights, and contingent on the NP having a collaborative arrangement in place with a physician [30].

We coded the "venue" column to determine whether or not the event was held in a clinical setting and the "description" column to determine the type of event (e.g. journal club, grand rounds, conference etc) using the coding strategy established by Fabbri et al [14]. This coding strategy initially comprised types of events described in the literature on industry-health professional interactions and then iteratively added keywords to account for all event types in the dataset [14].

### Analysis

To address the primary objective, the frequency and percentage of events were tallied overall and by company, and then broken down by whether the event was pain-related. Differences across companies in the latter were assessed using a chi square test or Fisher's exact test when the expected count was less than 5. Among the pain-related events, the frequency and percentage of events were compared across professional groups and then further broken down by event location and format. Total costs of events and costs for food and beverages (in Australian dollars) were summed overall and by professional group. Median costs per event were calculated overall and by professional group and presented with an Inter-Quartile Range (IQR). Differences across groups in median costs were assessed using the Kruskal-Wallace test. The frequency and percentage of pain-related events was reported for each of the 12 content foci variables and compared across professional groups using a chi square test or Fisher's exact test, as above. Statistical significance was established for omnibus tests at $p < .05$ and adjusted for post-hoc tests using the Bonferroni correction (i.e., $p < .05/3$ or $.017$). All analyses were undertaken in SAS (version 9.4).

## Results

In total, there were 10,690 events sponsored by these three opioid manufacturers across Australia in the 4-year observation period (out of a total 116,845 pharmaceutical industry-sponsored events). We focused our analysis on a comparison among three groups: Events with physicians-only; events with at least one nurse in attendance; and events including only nurses to facilitate comparison (Fig 1).

There were 10,006 events sponsored by opioid manufacturers for physicians only or events where at least one nurse was in attendance. Over a third of these events was sponsored by each of Mundipharma (n = 3824; 38%) and Janssen (n = 3672; 37%), and a quarter of the events were sponsored by CSL/bioCSL (n = 2510, 25%).

We identified 3,411 pain-related events sponsored by 3 companies: bioCSL/CSL (n = pain-related 15/2,510 total events; <1%); Janssen (n = 134/3,672; 4%); and Mundipharma (n = 3,262/3,824; 85%). The majority of pain-related events were multidisciplinary, with at least one nurse in attendance (1,964/3,411; 58%); 19 multidisciplinary events explicitly included nurse practitioners. Physicians were the only professionals in attendance at 38% of events (1,281/3,411); these events exclusively comprised physicians across specialties and all levels of training. Nurses were exclusively present at 5% of events (166/3,411), one of which included nurse practitioners only. Events with attendees from multiple professions were highly multidisciplinary and inclusive: 54% (1066/1964) included only physicians and nurses, while the remainder also included pharmacists (673/1964, 34%), physiotherapists (151/1964, 8%), psychologists (86/1964, 4%), occupational therapists (56/1964, 3%), social workers (39/1964, 2%), dieticians (22/1964, 2%), and various categories of administrative, health system, and research staff.

Table 2 details the characteristics of pain-related events. The three opioid manufacturers spent more than AUD 7.5 million on pain-related events; however, the median cost per attendee was nominal (approximately AUD 15). Food and beverage costs accounted for about a third of the total costs, at just over AUD 2.5 million overall. Physician-only events accounted for about half of the total spend (AUD $3,962,226, 52%). The most expensive event (at about AUD 350,000) was for physicians only, however physician-only events also had the largest range in event costs. Spending on nurse-only events was proportionate to the number of nurse-only events. In contrast to physician-only events, the most expensive nurse-only event was AUD 16,824, however, nurse-only events had the highest median cost per event at approximately AUD 930, which equalled to a median of approximately AUD 26 per attendee.

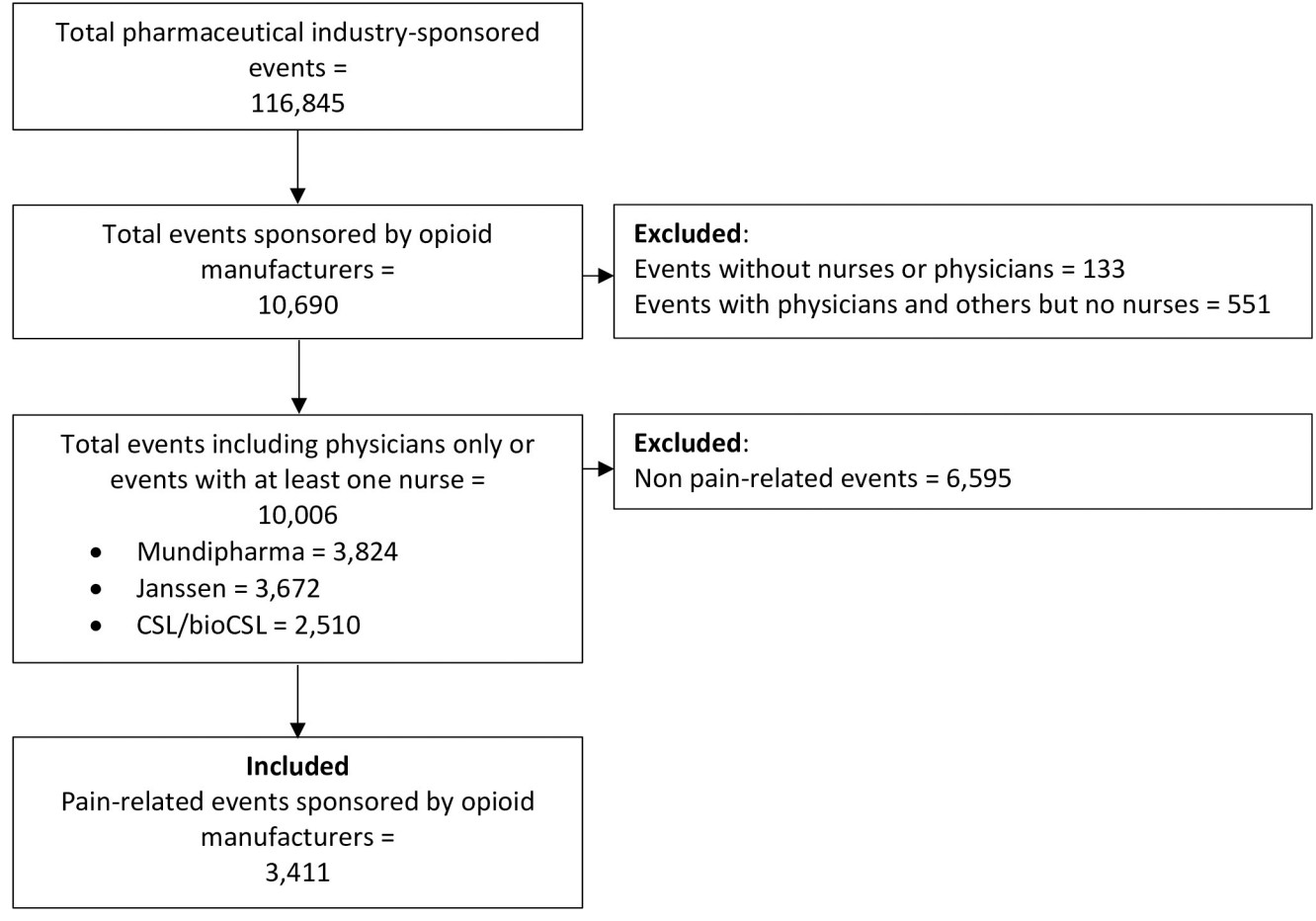

**Fig 1. Sampling flow diagram.**

More than half of all pain-related events took place in a clinical setting (2,067/3,411; 60%), which was also the case for events with physicians only (819/1,281; 64%) and those including at least one nurse (1,195/1,964; 61%). By contrast, pain-related nurse-only events occurred in a clinical setting at a significantly lower frequency (53/166; 32%; p < .001).

The most common format for pain-related events overall was a journal club (43%, 1,458/3,411), followed by approximately one quarter designated as a generic 'meeting' (24%, 827/3,411). Conferences and training were the least common, each making up 3% or less of the total events. Journal clubs and grand rounds were more commonly used for events including physicians compared to nurse-only events (p < .001), whereas nurse-only events more frequently had the format of a product-launch compared to the other event types (p < .001).

Table 2 provides a summary of the pain-related events according to the content focus and Table 3 provides illustrative examples of events with each content focus. For about a third of events (32%, 1,108/3,411), manufacturers did not report sufficient information to determine the content focus, mentioning only the event type and duration of the event (e.g. "journal club.– 1 hour duration," "departmental meeting– 2 hour duration"). For the remainder, the description of the event mentioned a topic (e.g. "Palliative care study day"), provided the title of a presentation (e.g. "Reducing the burden of persistent pain"), mentioned a particular product (e.g. "Series SR Hydromorphone"), or the focus could be inferred by the venue (e.g. aged care facility) or specialty of professionals in attendance (e.g. orthopaedic surgeons).

**Table 2. Characteristics of pain-related educational events sponsored by opioid manufacturers (N = 3,411).**

| Characteristic | Total pain-related N (%) | Prescribers[a] only N (%) | At least 1 nurse[b] N (%) | Nurses only N (%) | P-value[c] |
|---|---|---|---|---|---|
| Total number of events | 3,411 | 1,281 (38%) | 1,964 (58%) | 166 (5%) | |
| **Number of attendees** | | | | | |
| Total number of attendees (%) | 111,009 | 48,736 (43%) | 54,816 (49%) | 7,457 (7%) | |
| Median number of attendees (IQR) | 21 (15–31) | 20 (13–29)[ab] | 22 (15–31)[ac] | 35 (22–55)[bc] | < .001 |
| **Expenses** | | | | | |
| Total cost of events[d] | $7,608,754 | $3,962,226 (52%) | $3,340,841 (44%) | $305,687 (4%) | |
| Median cost per event (IQR) | $321 ($177-$2311) | $287 ($165-$2033)[ab] | $329 ($189–2403)[ac] | $930 ($257-$2594)[bc] | < .001 |
| Range of costs per event | $0[e]-$351,210 | $0-$351,210 | $14-$110,767 | $62-$16,824 | |
| Median cost/ median number of attendees per event | $15.27 | $14.37 | $14.95 | $26.56 | |
| Total cost of food and beverages | $2,477,132 | $943,003 (38%) | $1,382,896 (56%) | $151,234 (6%) | |
| Median cost of food and beverages per event (IQR) | $220 ($121-$887) | $180 ($98-$493)[a] | $245 ($136-$1046)[a] | $262 ($0-$1493) | < .001 |
| **Location[f]** | | | | | |
| Clinical setting[g] | 2,067 (61%) | 819 (64%)[a] | 1,195 (61%)[b] | 53 (32%)[ab] | < .001 |
| **Format[c]** | | | | | |
| Journal clubs | 1458 (43%) | 561 (44%)[b] | 877 (45%)[c] | 20 (12%)[bc] | < .001 |
| Meeting | 827 (24%) | 303 (24%) | 486 (25%) | 38 (23%) | = .713 |
| Grand rounds | 292 (9%) | 146 (11%)[ab] | 144 (7%)[ac] | 2 (1%)[bc] | < .001 |
| Product launch | 248 (7%) | 57 (4%)[ab] | 163 (8%)[ac] | 28 (17%)[bc] | < .001 |
| Conference | 103 (3%) | 63 (5%)[a] | 32 (2%)[ac] | 8 (5%)[c] | < .001 |
| Training | 23 (1%) | 5 (0%)[b] | 13 (1%)[c] | 5 (3%)[bc] | = .001 |
| **Content focus[h]** | | | | | |
| Chronic pain | 879 (26%) | 355 (28%) | 490 (25%) | 34 (20%) | = .059 |
| Cancer pain /palliative care | 606 (18%) | 144 (11%)[ab] | 408 (21%)[ac] | 54 (33%)[bc] | < .001 |
| Generic/unspecified pain | 515 (15%) | 180 (14%)[b] | 290 (15%)[c] | 45 (27%)[bc] | < .001 |
| Geriatrics | 274 (8%) | 97 (8%)[b] | 135 (7%)[c] | 42 (25%)[bc] | < .001 |
| Adverse effects | 95 (3%) | 30 (2%) | 65 (3%) | 0 (0%) | = .022 |
| Branded drug | 73 (2%) | 20 (2%)[b] | 39 (2%)[c] | 14 (8%)[bc] | < .001 |
| Non-pharmacological treatment | 33 (1%) | 27 (2%)[a] | 3 (0%)[ac] | 3 (2%)[c] | < .001 |
| Acute pain | 29 (1%) | 15 (1%) | 11 (1%) | 3 (2%) | = .070 |
| Addiction | 11 (0%) | 7 (1%) | 3 (0%) | 1 (1%) | = .126 |
| Orthopaedics | 12 (0%) | 9 (1%)[a] | 2 (0%)[a] | 1 (1%) | = .016 |
| Nerve pain | 3 (0%) | 1 (0%) | 2 (0%) | 0 (0%) | = .903 |

Abbreviations: IQR, interquartile range

Cost variables are reported in AUD

[a]Prescribers included physicians at all levels of training (n = 1,282 events) and nurse practitioners (n = 2 events)

[b]Events where at least one nurse was in attendance in addition to at least one physician, pharmacist, physiotherapist, psychologist, and/or allied health professionals

[c]The omnibus p-value is derived from a chi square test for frequency counts and a Kruskal-Wallace test for medians (p < .05). Differences between groups are indicated using matching letters (p < .017)

[d]Total cost of function includes food and beverages and/or venue and audio-visual hire, speaker honoraria, speaker and attendee airfare and accommodation, parking, and meeting sponsorship

[e]Company reported $0 for 2 events, stating "Solely sponsored honorarium for international speaker" and "No hospitality or honoraria provided."

[f]Reported percentages for Location and Format are column percentages which do not add to 100% as some events had insufficient detail to code

[g]Non-clinical settings largely included restaurants, hotels, and convention centres

[h]Reported percentages for Content focus are column percentages and may add to >100% as events could be coded for multiple foci

For those where content focus could be determined, events focused on chronic pain (26%, 879/3,411), followed by cancer pain and palliative care (18%, 606/3,411), and then generic or unspecified references to pain such as "A new approach to pain management," or "Pain

**Table 3. Illustrative examples of pain-related events by content focus[a].**

| Content focus | Company | Date | Description of function | Format/ duration | Venue | No. Attendees (Type) | Total cost of function |
|---|---|---|---|---|---|---|---|
| **Chronic pain** | Janssen | 2012-Jun | Part 1 of 2 How to treat chronic pain using a biopsychosocial/ multi-disciplinary approach | Training/ 3 hr | Hotel | 23 (General Practitioners, Registrars) | $1,958 |
| **Cancer** | Janssen | 2013-Jun | Cancer and Palliative Care Update | Meeting sponsorship/ 6 hr | Aged care facility | 23 (Nurses) | $166 |
| **Generic pain** | Mundipharma Pty Ltd | 2012-Apr | A new approach to pain management (Product Launch Meeting) | Dinner meeting/ 1.5 hr | Restaurant | 40 (General Practitioners, Nurses, Pharmacists) | $3,470 |
| **Geriatrics** | Mundipharma Pty Ltd | 2011-Nov | Pain in the elderly meeting the challenges | Dinner meeting/ 2 hr | Conference centre | 29 (Nurses) | $2,314 |
| **Adverse effects** | Mundipharma Pty Ltd | 2015-May | CME Meeting: Chronic Pain & Constipation. Total of 2 RACGP (Cat 2) QI & CPD points | Dinner meeting/ 2 hr | Restaurant | 24 (General Practitioners, Nurses) | $4,540 |
| **Branded drug** | Mundipharma Pty Ltd | 2013-Aug | Meeting: Presenting TARGIN© tablets and NORSPAN© patch | Dinner meeting/ 1.5 hr | Restaurant | 47 (Nurses, Pharmacists) | $4,266 |
| **Non-pharmacological** | Mundipharma Pty Ltd | 2012-Apr | Pain Management: A Psychosocial Perspective | Dinner meeting/2 hr | Restaurant | 88 (Nurses) | $4,375 |
| **Acute pain** | Mundipharma Pty Ltd | 2013-May | "Perioperative Pain Management: Acute Pain Service update", Mundipharma sponsored St John of God Hospital and had no involvement in inviting attendees or organising the educational component | Meeting sponsorship/ 1.5 hr | Hospital | 60 (Anaesthetists, Nurses, Registrars, Nurse Practitioners, Orthopaedic Surgeons, Accident & Emergency Doctors) | $1,887 |
| **Addiction** | Janssen | 2012-Aug | Chronic Pain Treatment Option and Opioid Addiction | Lunch meeting/ 2 hr | Hospital | 18 (Nurses) | $63 |
| **Orthopaedics** | Janssen | 2012-Oct | Hips, Shark Bites and Drugs<br><br>This event was organised by Australian Orthopaedic Nurses' Association and Janssen was not responsible for inviting the attendees or organising the educational content | Dinner meeting/ 2 hr | Restaurant | 19 (Consultant (Geriatrics, Rehabilitation Medicine), General Practitioner) | $1,045 |
| **Nerve** | Janssen | 2013-Apr | Neuropathic pain post surgery | Finger food and journal club/ 1 hr | Hospital | 15 (Consultant (Pain Management, Rehabilitation Medicine, Rehabilitation Physician, Rheumatology), Nurse, Psychiatrist, Psychologist (Pain Management), Registrar) | $156 |

CME = continuing medical education; RACGP = The Royal Australian College of General Practitioners; QI = quality improvement; CPD = continuing professional development

[a]Event data is taken verbatim from company reports submitted to Medicines Australia

Medicine refresher course," or "Global day of pain" (15%, 515/3,411). All other types of specific pain (acute, nerve, orthopaedic) were each the focus of 1% of events or less. Less than one in 10 events related to pain among geriatric populations, including the mentioning of older adults, or pain as a co-morbidity for conditions such as dementia (8%, 274/3,411). Only 3% (95/3,411) of all events mentioned an adverse event or safety considerations. Of these, 87% (83/95) specifically mentioned constipation. Events referenced a brand name drug in 73 cases (2%).

The content focus varied across professional groups, though there was no significant difference among groups regarding events focused on chronic pain. Events focused on cancer pain or palliative care were more common for nurse-only audiences (33%) than physician-only (11%) or multidisciplinary (21%) ones (p < .001). A similar pattern was found for events focused on generic/unspecified pain topics (27% of nurse-only vs 14% of physician-only and 15% of multidisciplinary; p < .001). A significantly larger proportion of nurse-only events also focused on topics related to geriatric populations and brand-name drugs than physician only or mixed events (each p < .001).

## Discussion

Over a four-year period, over 111,000 Australian physicians, nurses, and other health professionals attended 3,411 educational events focused on topics and treatments related to pain that were sponsored by 3 opioid manufacturers. Mundipharma, a privately held company, which is owned by the Sackler family, the owners of Purdue Pharma, sponsored 96% of the educational events in this analysis. Though details regarding the event's focus were missing for about 1/3 of events, chronic pain was a prominent focus for all audiences, which is consistent with Purdue's promotion of oxycodone in the United States [7]. In contrast, events focused on cancer pain and palliative care, for which opioids are indicated, were less frequently a focus of event content, particularly for physicians. Thus, industry-sponsored educational events may have focussed on content that was not evidence-based (i.e. the use of opioids for the long-term treatment of chronic non-cancer pain) and routinely included healthcare teams, including non-prescribing nurses. In examining the role of pharmaceutical industry promotion as a precursor to the inappropriate and harmful prescribing of medications more generally, these data suggest that the "educational" nature of sponsored events should be called into question and that policy efforts to ensure the quality use of medicines should be inclusive of non-prescribers and nurses in particular.

These data provide additional insights into the nature of promotion of brand name, prescription opioids through events targeted at health professionals, which appear consistent with the activities of manufacturers' multi-national parent companies and subsidiaries globally [7, 9, 17]. However, these data represent what is likely a fraction of total promotional spending: 2012 data from the United States found that sponsorship of educational events accounted for 8% of total marketing expenditure, while detailing (face-to-face visits from sales representatives) accounted for 55% [31]. That said, this dataset is unique in that expenditures included sponsorship, hospitality, speakers' fees, and travel reimbursement, so may account for spending beyond what is typically considered "event sponsorship."

These data also provide only a snapshot of the promotion of opioids by three manufacturers of prescription, brand name opioids who were then-members of Medicines Australia (and thus subject to voluntary reporting requirements); these data are not representative of all opioid manufacturers including non-member companies, manufacturers of generic drugs, nor others involved in the distribution and dispensing of opioid medications. In the United States, opioid manufacturers and distributors are currently faced with more than 2000 lawsuits alleging that opioid marketing was false and misleading in downplaying the harms and overstating the benefits [17]. While Purdue Pharma and the other top opioid manufacturers by total sales (including Purdue Pharma, Johnson & Johnson's Janssen division, Insys, Mylan, and Depomed) have received the greatest public, legal and regulatory scrutiny, federal and state governments have brought legal action against manufacturers of generic opioids (e.g. Endo International, Mallinckrodt Pharmaceuticals and Teva Pharmaceutical Industries), the major distributors (e.g. McKesson, Cardinal Health, and AmerisourceBergen), and a corporate

pharmacy chain (Walmart Inc.) [32], suggesting that the promotional and commercial origins of the opioid epidemic are highly complex [17, 33, 34].

This analysis provides an Australian perspective on the promotion of opioids to health professionals, adding to analyses of United States data [10–12]. Coinciding with the study period, there was rapid uptake of Mundipharma's oxycodone/naloxone controlled-release product in Australia with promotion focused on its beneficial effects on opioid-induced constipation [35]. An analysis of dispensing claims from 2006–2016 found that, following approval of the 5 mg oxycodone/naloxone controlled-release formulation for subsidy under the federal government Pharmaceutical Benefits Scheme, dispensing claims grew 1.6-fold for oxycodone controlled-release dispensing, driven predominantly by rapid uptake of the low strength (≤5 mg) oxycodone/naloxone controlled-release formulation [35]. Uptake of the drug was greater than expected if it were substituting the single-ingredient oxycodone controlled-release, suggesting instead, an expansion of the oxycodone controlled-release market [35]. Uptake was greatest among older individuals initiating opioid treatment, despite recommendations against initiation with a long-acting formulation due to increased risk of adverse effects such as accidental overdose and respiratory depression [35].

In the context of the literature, these results thus suggest that industry-sponsored education may further promotional aims. In the small minority of cases, such as the 248 "product launch" events (e.g. "3 course meal with alcohol/non- alcoholic beverages, Janssen Educational Event: Series SR Hydromorphone. A once a day oral opioid for chronic pain management") or the 73 events that promoted a brand name drug (e.g. Dinner 3 course, "Presenting TARGIN© Tablets Launch Meeting", Mundipharma has sponsored a General Practitioner Network and had no involvement in inviting attendees or organising the educational component. 2 hour duration"), the promotional intent was explicit. Where detailed, these events (sponsored by Mundipharma) described dinner events where the company's new brand name product was presented as a solution for chronic pain and opioid-induced constipation (e.g. Dinner 3 course, alcohol provided, CME Meeting: Chronic Pain & Constipation (with NORSPAN© patch). 2 hour duration"), which is consistent with the company's broader promotional campaigns in Australia during the study period [35]. These findings are concerning given that opioid manufacturers in several countries are facing legal action for misleading, inaccurate, and harmful advertising. For example, in 2019, the TGA, Australia's medicine regulator, fined Mundipharma AUD 302,400 over promotion of its oxycodone/naloxone controlled-release product as it presented the product as a core component of the multi-modal management of chronic non-cancer pain [36].

This analysis suggests that opioid promotion via sponsored educational events extends beyond physicians to multidisciplinary teams and also specifically, to nurses. Compared with pharmaceutical industry sponsored educational events in general, events targeted at multidisciplinary teams were over represented among pain-related events sponsored by opioid manufacturers (57% vs 40% overall); nurses were specifically targeted at a similar proportion (5.2% vs 4.8% overall) [22]. Although a majority of multidisciplinary pain-related events (54%) included physicians and nurses only, many were highly multidisciplinary, including a wide range of health professionals and members of the healthcare team, suggesting that promotional efforts are inclusive. Initiatives such as education or policy designed to promote the safe and quality use of opioid medications should adopt a similarly multidisciplinary and inclusive approach.

Events, including product launches, were also targeted specifically at nurses without prescriptive authority, indicating that non-prescribers are a target audience for pharmaceutical industry sponsorship beyond being collateral attendees at physician-focused events. Nurse-only events were more often held outside of clinical settings, and nurses were the intended

audience for events focussed on branded controlled-release formulations, cancer pain, and geriatric populations. These findings are consistent with literature that suggests nurses frequently exert influence over treatment decision making and play a role in medication compliance [37, 38]. The extent and impact of opioid promotion to nurses across healthcare settings remains underexplored and is an important avenue for future research. For example, following the study period, Mundipharma launched educational campaigns designed to facilitate their implementation in long term care settings (see "PAN.AC.EA," or, "Pain Advocacy Nurse in Aged Care for Education & Assessment [39]).

## Policy implications

Despite the routine attendance of nurses and other non-physician clinicians at industry-sponsored events, there is little policy guidance or regulation of these relationships. Non-prescribing registered nurses, pharmacists and other allied health professionals are covered by the Medicines Australia and European Federation of Pharmaceutical Industries and Associations (EPFIA) Codes of Conduct, both of which represent a form of industry self-regulation. While comprehensive in terms of the recipients covered, these self-regulatory approaches fall short of comprehensive disclosure of all forms of industry payments (i.e. frequent exclusion of food and beverages, drug samples) and apply only to members of these trade associations, who disclose on a voluntary basis [40, 41]. Legislation such as the United States Physicians Payments Sunshine Act has addressed these limitations in requiring comprehensive, standardized, and enforceable reporting. Though several categories of nurses with prescribing authority (e.g. nurse practitioners, clinical nurse specialists) will be covered by this legislation beginning in 2022 [42], policymakers should consider enacting transparency legislation that covers all registered health professionals. While transparency is necessary, it is insufficient to mitigate or prevent harms associated with industry relationships. Thus, professional associations, regulators and accrediting bodies for nurses, pharmacists and allied health professionals need to develop guidance related to ethical interaction with industry and management of conflicts of interest. This will likely be most effective in practice if developed and implemented in a way that is inclusive of all members of the healthcare team.

Regulators and industry trade associations have attempted to distinguish between companies' "promotional" and "non-promotional" activities [26, 43], with health professional education and disease awareness campaigns typically included in the latter. Regulators should instead consider regulating promotional campaigns, comprised of sales, marketing, advertising, promotion, research and educational activities, as a whole to restrict the promotion of products that pose a high risk of harm. However, this also raises the question of who should be responsible for delivering education about pharmaceuticals to healthcare professionals and suggests pragmatic challenges as industry-independent educational interventions are in no way comparable in terms of resources, reach and duration to industry-sponsored educational events. Two Australian trials examined the effects of independent chronic pain management education programs among general practitioners and found no significant effect on prescribing habits or intent [44, 45]. However, a systematic review and meta-analysis of interventions to improve prescribing practices for chronic non-cancer pain found very low-certainty evidence that education could indirectly improve prescribing practices through increased uptake of treatment agreements, urine drug testing, and use of prescription drug monitoring programs [46]. Thus, the question is perhaps not how to deliver 'counter' education, which may not be effective on its own in mitigating the harms associated with industry-sponsored education, but how it might be incorporated into multi-faceted [46], multidisciplinary, and structural efforts to improve the quality use of medicines and reduce opioid-related harms.

## Strengths and limitations

This was a cross-sectional, descriptive analysis of data that were publicly reported by pharmaceutical companies; thus, we were restricted by the format in which data were reported and could not verify their accuracy or completeness. Further, there was a high degree of variability in terms of the specificity and comprehensiveness of reporting with insufficient detail to determine the content focus of about 1/3 of sampled events. We iteratively coded data using keywords according to each variable of interest and due to the variability of the underlying reports, we may have missed some synonyms or categories. We may have overestimated the number of events that were "pain-related" and sponsored by Mundipharma in assuming that events were pain-related in the absence of details about the event content or attendee specialty. However, given that Mundipharma's product portfolio is dominated by pain-related medications and the promotion of opioid medications by Mundipharma companies both overtly and covertly is well-documented, we believe this was a reasonable assumption [9]. The original reports provided the total number of attendees and listed the professional status of attendees (i.e. physician, registrar, nurse, pharmacist etc); however, the reports did not specify the number of attendees by professional status. Thus, we could only make comparisons based on whether certain professionals were present or absent. Despite these limitations, this analysis is the first to describe the content of educational events sponsored by opioid manufacturers and also the nature of opioid-related promotion to nurses.

## Conclusions

Opioid manufacturers globally are being called to account for their role in seeding and exacerbating epidemics of opioid-related morbidity and mortality through the aggressive promotion of prescription opioids. Similar scrutiny should be extended to industry sponsorship and delivery of health professional "education" related to opioids and pain management. Pain-related educational events for health professionals are inclusive of multidisciplinary teams and nurses working across healthcare settings and are focused on the treatment of chronic pain, including in vulnerable populations. The content of sponsored education echoed promotional campaigns during the study period, thus, regulators should consider placing restrictions on promotional campaigns rather than drawing false distinctions between "advertising" and "education."

## Supporting information

**S1 Table. Variables of interest and keyword search strategy for sponsored educational events.**
(DOCX)

**S2 Table. Coding manual for content analysis.**
(DOCX)

## Author Contributions

**Conceptualization:** Quinn Grundy, Emily A. Karanges.

**Data curation:** Quinn Grundy, Sasha Mazzarello.

**Formal analysis:** Sarah Brennenstuhl.

**Funding acquisition:** Quinn Grundy.

**Investigation:** Sasha Mazzarello, Emily A. Karanges.

**Methodology:** Quinn Grundy, Emily A. Karanges.

**Project administration:** Sasha Mazzarello.

**Software:** Sarah Brennenstuhl.

**Supervision:** Quinn Grundy.

**Writing – original draft:** Quinn Grundy.

**Writing – review & editing:** Sasha Mazzarello, Sarah Brennenstuhl, Emily A. Karanges.

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
