## [Decision Letter · Decision Letter 0]

23 Dec 2020

PONE-D-20-20905

A comparison of educational events for physicians and nurses in Australia sponsored by opioid manufacturers

PLOS ONE

Dear Dr. Grundy,

Thank you for submitting your manuscript to PLOS ONE. After careful consideration, we feel that it has merit but does not fully meet PLOS ONE’s publication criteria as it currently stands. Therefore, we invite you to submit a revised version of the manuscript that addresses the points raised during the review process.

A premise for acceptance is adequate consideration of the issues raised regarding data coding/analysis and characteristics of events.

We look forward to receiving your revised manuscript.

Kind regards,

Tim Mathes

Academic Editor

PLOS ONE

Journal Requirements:

'The authors have no conflicts of interest. Quinn Grundy currently serves as Academic Editor for PLOS ONE.'

a. Please confirm that this does not alter your adherence to all PLOS ONE policies on sharing data and materials, by including the following statement: "This does not alter our adherence to  PLOS ONE policies on sharing data and materials.” (as detailed online in our guide for authors http://journals.plos.org/plosone/s/competing-interests).  If there are restrictions on sharing of data and/or materials, please state these.

Please note that we cannot proceed with consideration of your article until this information has been declared.

Reviewers' comments:

Reviewer's Responses to Questions

**Comments to the Author**

1. Is the manuscript technically sound, and do the data support the conclusions?

Reviewer #1: Yes

Reviewer #2: Yes

Reviewer #3: Yes

2. Has the statistical analysis been performed appropriately and rigorously? 

Reviewer #1: Yes

Reviewer #2: Yes

Reviewer #3: Yes

3. Have the authors made all data underlying the findings in their manuscript fully available?

Reviewer #1: Yes

Reviewer #2: Yes

Reviewer #3: Yes

4. Is the manuscript presented in an intelligible fashion and written in standard English?

Reviewer #1: Yes

Reviewer #2: Yes

Reviewer #3: Yes

5. Review Comments to the Author

Reviewer #1: This study addressed an important issue of industry promotional activity in the medical field; clarifying the current situation among opioid manufacturer-sponsored educational events in Australia. The study also focused on the differences between pain-related educational events for physician and those for nurses. This is a cross sectional study with numerous data, the manuscript is well structured and the methodology and analysis seems to be appropriate.

I have added some following comments.

#Method

The authors describes in the "Coding" section, "We coded all Mundipharma events as "pain-related" unless the event explicitly stated otherwise, given the company's limited product profile"

I understand that mundipharma's main products are pain-related drugs, but the profile may vary from country to country. Do you have the data to support this?

#Result

Was there any difference in the details of the events between the three companies?

#Discussion

While it is conceivable that pharmaceutical companies may do more promotional activities when there are competing drugs, opioids are promoted with a different aim to increase the target population, which has had the unfortunate consequence of inappropriate prescriptions. Most events in the present study were held by mundipharma. Is the majority of pain-related prescriptions in Australia by mundipharma? If so, do you see these efforts of mundipharma as part of promotion to increase prescribing through education rather than competition with competing companies?

Reviewer #2: General comments

This article addresses a hugely important public health issue. It is well written and insightful. Thank you for the opportunity to review it. It does a great job of extending the filed of research on industry payments beyond the US as well as beyond physicians. The findings and discussion are very clear. I only have a few questions regarding the introduction and methods.

Introduction

It would be useful to see a bit more justification of why Australia is a key case to study the behaviour of opioid manufacturers.

In addition, it would be helpful to see more background on why nurses are key to target in opiod drug promotion in general and specifically in Australia. Is there any data on the sThis would help international readers situate the article in a broader context.

Beyond payment reporting, how are drug company interactions with nurses regulated in Australia, i.e. professional associations, hospitals, government regulation?

Methods - sampling

Could you provide a bit more context on the opioid product portfolios of the three companies. Are there any significant differences between them? Equally important, to what extent, in your evaluation, do these companies representat opioid drug manufacturers more broadly? It would be helpful to see some commentary on these issues to be able to contextualise the findings. Some of the information on companies could be placed in the introduction; while you mention some of it in the discussion I believe it would be better to start off with a bigger picture of the opioid sector.

Methods - coding

Please could you provide a bit more information on event descriptions? How extensive were they? Were there any differences in the content or style of descriptions between the included companies? Having been involved in coding of desriptions of payments to patient organisations in the UK I am aware that the specificity and comprehensiveness of payment descriptions can vary considerably, which may have implications for the validity and reliability of coding. It would be good to see some reflection on these points.

Could you provide more background on the event types categories? Were these deductive or deductive? If they were taken from manufacturers do they have any standardised descriptions? IF the categories were inductive please explain how they were derived.

Methods - analysis

Did you consider conducting a separate set of analyses for each company and perhaps trying to interpret it in the context of its product portfolio?

Reviewer #3: In this cross-sectional descriptive analysis of pharmaceutical company reports, Dr. Grundy et al examined opioid-related industry educational events for physicians, nurses, and other healthcare professionals from 2011-2015 in Australia.

This is a very important issue, and there are not many data sources that capture industry promotion to non-prescribers (e.g. RNs or pharmacists). Dr. Grundy et al use industry reports, which seem to be limited in details (e.g. not a ton of information presented about the content of these events, and no available breakdown about how many physicians, nurses, or other professionals were at each event).

My main suggestion is that the research question could be more crisply articulated; a lot of characteristics of these events are presented, but the authors could be more explicit about the research question, whether these results are surprising, and what are the direct clinical or policy implications of these data. I have some additional major and minor comments/suggestions below.

Major comments

Methods:

- “Coding” para 1 – the authors state “We coded all Mundipharma events as “pain-related” unless the event explicitly stated otherwise, given the company’s limited product profile.” Looking at their website mundipharma.com.au, it seems they also make eye drops, naloxone, fluticasone/eformoterol inhaler, and a bunch of oncology drugs, as well as buprenorphine (which is not included per Methods) – so I’m not sure this is a valid assumption. The name-brand inhaler and oncology agents could be heavily marketed, as could naloxone…

- I think the decision not to separately code NPs needs a little more justification, since the paper’s core objective is describing the difference between prescriber-oriented and nursing-oriented educational events. If some of the nurses at “events with at least one nurse” were actually NPs, wouldn’t those events be miscategorized in the analysis (i.e. they were actually prescriber-only)?

- It is a limitation that the data does not specify how many of each profession were present at each event (this is stated in the discussion, but may be helpful to include earlier, in the methods). I think the authors did they best they could to offset this, by examining physician-only and nursing-only events separately, but nursing-only events are few, and it’s hard to know what to make of the mixed-profession events, without more context or breakdown of professions in attendance

Results:

- There are not very many RN-only events, and I don’t have total clarity about the mixed MD/nursing events (defined as “at least one nurse”; is it just a few nurses or substantial representation? Are we sure that those nurses are non-prescribers, and not NPs?)

- I also am not sure that these events (>40% journal clubs) are where most of the educational marketing spending is occurring (just $7M over 5 years); does Australia allow door-to-door “educational” detailing in clinical settings, and is it possible that constitutes more of the marketing budget?

Discussion:

- I appreciated the point at the end of Para 1 – cancer pain is a stronger indication for opioids, but less featured in educational events. In my mind, the 2 most important conclusions to highlight from these data are that 1) opioid marketing to non-prescribers must be considered, and 2) there is still significant opioid marketing around chronic pain, for which it is not evidence-based

- I think the authors’ conclusion that educational activities may still be promotional in nature is very astute. It could be more clearly supported by the data (e.g. more explicit examples of the case of oxycodone/naloxone and constipation)

Minor comments:

Intro:

- I wasn’t familiar with bioCSL/CSL or Mundipharma – it may be worth providing a little context about these companies in the intro (I didn’t realize until the discussion that Mundipharma is owned by the Sackler family, e.g.)

- Abstract results – “the majority…were focused on chronic pain (26%)” – I would instead say that chronic pain was the most common topic, followed by XYZ

- Would be interesting to hear what other multidisciplinary groups were represented at these educational events – e.g. pharmacists? Hospital administrators? PAs and NPs?

- Intro line 5 – what is the unit when you say “the rate of death increased from 1 in 1999 to 4.4 in 2016”?

- The intro doesn’t mention Australia until page 3 – may frame the discussion for Australian data earlier

- Intro page 2 line 4 – would also include industry-sponsored detailing visits and meals as a very common “educational” activity

- Typo intro p2 para 2 – “analyses…suggests”

- End of page 5 – might clarify or flip this sentence (payments associated w/ increased mortality, rather than the other way around)

- This sentence could be clarified: “In Australia, nurses attended 40% of pharmaceutical-industry sponsored educational events at twice the rate of primary care physicians [15].”

Methods:

- Would include a sentence of explanation why you excluded meds for breakthrough cancer pain

Results:

- Para 3 (describing the cost range between events) was a little hard to follow

- Table 2 – would include median cost per attendee. Also curious what the non-clinical settings were

Discussion:

- Para 2 – what do you mean by oxycodone/naloxone’s “subsidy”? I wonder if increased uptake among older adults was due to increased constipation in this population

- Para 4 suggests that opioid events targeted more multidisciplinary groups, but the same # of nurses. Was it other professions driving the multidisciplinary nature of opioid events (eg pharmacists?)

- Would be cautious of the statement that “nurses attended product lunches more frequently than groups including physicians,” since there were 248 such events for physicians-only and 28 product launches for nurses-only

- Para 5 and 6 may overstep the data somewhat – not sure it can be extrapolated from the differences in Table 2 that “this type of promotion may reflect the influence of nursing in community, home and long-term care settings and create risk for misuse of opioids in vulnerable populations”. Could it be argued that nursing education improves safety or control of hospice & cancer pain, or that some of these Mundipharma trainings were in reference to rescue use of Narcan (naloxone)?

- Is it really true that non-industry education about opioids has not been effective? Is there a review (eg Cochrane) that you could cite?

- I appreciate the point that even pharmaceutical “educational” events should be regulated as promotion. There are questions that arise as to who would deliver education if Pharma did not (e.g. without these industry activities, would it take longer for an effective new drug to reach prescribers & patients – even if that info is favorably biased)

6. PLOS authors have the option to publish the peer review history of their article (what does this mean?). If published, this will include your full peer review and any attached files.

Reviewer #1: **Yes: **Hiroaki Saito

Reviewer #2: No

Reviewer #3: No

---

## [Author Response · Author response to Decision Letter 0]

25 Jan 2021

Please see the attached response to reviewer documents in which all comments and responses are tabled.

---

## [Decision Letter · Decision Letter 1]

17 Feb 2021

PONE-D-20-20905R1

A comparison of educational events for physicians and nurses in Australia sponsored by opioid manufacturers

PLOS ONE

Dear Dr. Grundy,

Thank you for submitting your manuscript to PLOS ONE. After careful consideration, we feel that it has merit but does not fully meet PLOS ONE’s publication criteria as it currently stands. Therefore, we invite you to submit a revised version of the manuscript that addresses the points raised during the review process.

We look forward to receiving your revised manuscript.

Kind regards,

Tim Mathes

Academic Editor

PLOS ONE

Reviewers' comments:

Reviewer's Responses to Questions

**Comments to the Author**

1. If the authors have adequately addressed your comments raised in a previous round of review and you feel that this manuscript is now acceptable for publication, you may indicate that here to bypass the “Comments to the Author” section, enter your conflict of interest statement in the “Confidential to Editor” section, and submit your "Accept" recommendation.

Reviewer #1: All comments have been addressed

Reviewer #2: All comments have been addressed

Reviewer #3: All comments have been addressed

2. Is the manuscript technically sound, and do the data support the conclusions?

Reviewer #1: Yes

Reviewer #2: Yes

Reviewer #3: Yes

3. Has the statistical analysis been performed appropriately and rigorously? 

Reviewer #1: Yes

Reviewer #2: Yes

Reviewer #3: Yes

4. Have the authors made all data underlying the findings in their manuscript fully available?

Reviewer #1: Yes

Reviewer #2: Yes

Reviewer #3: Yes

5. Is the manuscript presented in an intelligible fashion and written in standard English?

Reviewer #1: Yes

Reviewer #2: Yes

Reviewer #3: Yes

6. Review Comments to the Author

Reviewer #1: The authors have responded to my comments adequately and the revised manuscript has been refined.

Regarding the coding section, I understood the situation surrounding Mundipharma's products in Australia and recognized that the limitations of this study were appropriately mentioned. Thank you also for adding the discussion on promotion appropriately.

One point, probably a simple mistake, but in Table 2, under Median cost/median number of attendees per event, should the Nurses only column be $2656 -> $26.56? Should median cost per events for Nurses only be also modified(from $930 to $909)?

Reviewer #2: Thank you for the opportunity to review the revised version of this manuscript as well as the Authors’ responses. I am very happy with how my original comments were addressed. I appreciate your very detailed and meticulous responses. Overall, I believe this paper will be a very important and timely contribution to the literature. Congratulations on completing this important piece of research.

I only have a few additional minor comments relating to the discussion.

To set the findings in a broader research context I would also suggest mentioning that nurses are have been covered by the EFPIA’s Disclosure Code (subsequently subsumed under the EFPIA Code) but the incompleteness of industry disclosures in Europe seem to be the main factor behind the lack of research on this issue. On the other hand, nurses will only be covered by Open Payments starting from 2021. So your research could be very valuable in terms of helping to set the agenda in these other jurisdictions, particularly the US and in relation to the opioid crisis. I think it would be useful to bring up these contexts.

Given the importance of your findings I’d like to have seen at least a few policy recommendations outlined in the Discussion. In particular, I find the lack of official guidelines from government or professional bodies very surprising and concerning as it appears that - in light of your findings - that nurses’ interactions with the industry are governed exclusively by the industry’s self-regulation. If this is the case then I believe it does amount to an important policy point.

Reviewer #3: The authors were insightful and thorough in their response to reviewers' comments, all of which they have addressed within the constraints of this unique and novel dataset. Limitations of the data (e.g., cannot determine the proportion of nurses at each educational event), have been well-described in the discussion. The revised intro and conclusion and extremely compelling and frame the importance of this issue for the reader.

Minor comments:

Abstract discussion:

- The authors state, “Regulators should consider the validity of distinguishing between pharmaceutical companies’ “promotional” and “non-promotional” activities.” In order to link this sentence to the results presented, could precede it with a sentence such as: “Chronic pain was the most common event topic, despite lack of evidence that opioids improve outcomes for chronic non-cancer pain. Regulators should consider…”

Table 2: “nurses only” column: $2656 instead of $26.56

7. PLOS authors have the option to publish the peer review history of their article (what does this mean?). If published, this will include your full peer review and any attached files.

Reviewer #1: **Yes: **Hiroaki Saito

Reviewer #2: No

Reviewer #3: No

---

## [Editor Report · Decision Letter 2]

23 Feb 2021

A comparison of educational events for physicians and nurses in Australia sponsored by opioid manufacturers

PONE-D-20-20905R2

Dear Dr. Grundy,

We’re pleased to inform you that your manuscript has been judged scientifically suitable for publication and will be formally accepted for publication once it meets all outstanding technical requirements.

Kind regards,

Tim Mathes

Academic Editor

PLOS ONE
---

## [Editor Report · Acceptance letter]

9 Mar 2021

PONE-D-20-20905R2 

A comparison of educational events for physicians and nurses in Australia sponsored by opioid manufacturers 

Dear Dr. Grundy:

I'm pleased to inform you that your manuscript has been deemed suitable for publication in PLOS ONE. Congratulations! Your manuscript is now with our production department. 

Kind regards, 

on behalf of

Dr. Tim Mathes 

Academic Editor

PLOS ONE